# Elevated Baseline Serum PD-L1 Level May Predict Poor Outcomes from Breast Cancer in African-American and Hispanic Women

**DOI:** 10.3390/jcm11020283

**Published:** 2022-01-06

**Authors:** Yanyuan Wu, Pranabananda Dutta, Sheilah Clayton, Amaya McCloud, Jaydutt V. Vadgama

**Affiliations:** 1Division of Cancer Research and Training, Department of Medicine, Charles R. Drew University of Medicine and Science, Los Angeles, CA 90059, USA; pranabandutta@cdrewu.edu (P.D.); smclayton1@yahoo.com (S.C.); amccloud@ucsd.edu (A.M.); 2Jonsson Comprehensive Cancer Center, David Geffen School of Medicine, University of California at Los Angeles, Los Angeles, CA 90095, USA

**Keywords:** PD-L1, TNBC, survival, CD44, PTEN

## Abstract

Background: The therapeutic targeting of PD-1/PD-L1 has shown clinical efficacy in treating metastatic breast cancer. We investigated the clinical significance of measuring serum PD-L1 levels in African-American and Hispanic women with breast cancer. Methods: PD-L1 levels were measured with the ELISA method from the serum samples of 244 African-Americans and Hispanics with breast cancer and 155 women without cancers. The levels of INFα2 and TNFα were measured with a Luminex multiplex assay. The protein levels of pAkt and CD44/CD24 in tumor cells were tested with immunohistochemistry analysis. Cox regression was used to assess the predicting role of serum PD-L1 for disease-free survival (DFS). Results: PD-L1 levels were significantly elevated in breast cancer cases compared to non-cancer cases. The high PD-L1 levels were associated with HER2-positive and triple-negative breast cancer. PD-L1 level independently predicted DFS in both African-American and Hispanic women. The evaluated PD-L1 level was found to be associated with high IFNα2 and TNFα in breast cancer patients. Conclusions: PD-L1 serum levels can predict DFS in African American and Hispanic women with breast cancer. Furthermore, a high level of PD-L1 is more likely to be associated with tumor loss PTEN and the activation of Akt or with breast cancer cells expressing CD44high/CD24low. Further validation studies are needed to determine if PD-L1 could serve as a biomarker for patient selection for anti-PD-L1 therapy and assess treatment outcomes.

## 1. Introduction

Breast cancer is the most common cancer among women (excluding skin cancers) and is the second leading cause of cancer death among women in the United States (US) [1]. Compared to different types of breast cancers, triple-negative breast cancer (TNBC) is a complex and highly aggressive subtype of breast cancer [2,3]. TNBC constitutes from 10% to 20% of all breast tumors and is characterized by a lack of expression of estrogen, progesterone, and human epidermal growth factor receptor 2 (HER2) receptors, thereby making it difficult to treat [4]. Among the various ethnic groups, African-Americans, especially younger African-Americans, are more likely to have TNBC, which considerably contributes to increased mortality and cancer health disparities in the US [3,5]. Another aggressive type of breast cancer is HER2-positive (HER2+) breast cancer. Until the discovery and use of trastuzumab for HER2+ breast cancer treatment, patients with HER2+ tumors had inferior disease outcomes [6,7]. However, almost 52% of HER2+ patients fail trastuzumab treatment, leading to disease progression [8]. Therefore, novel therapeutic strategies are needed to improve patients’ management of these types of breast cancers.

The programmed death receptor-1 (PD-1) and its ligand, programmed cell death-ligand 1 (PD-L1), are increasingly recognized as powerful targets to enhance tumor-directed cytotoxic T-cell function. The PD-1 is expressed on the surface of activated T, B, and natural killer (NK) cells [9], and it can interact with its two ligands, PD-L1 (also called B7H1/CD274) and programmed cell death-ligand 2 (PD-L2, or B7DC/CD273) [10]. PD-L1 is expressed on dendritic cells (DCs), macrophages, tumors, and immune cells of the tumor microenvironment, i.e., on stromal tumor-associated macrophages and lymphocytes [11]. The interaction of PD-L1/PD-1 on activated T-cells impedes T-cell function and promotes CD4+ T cells’ differentiation into regulatory T-cells (Tregs), eventually protecting the tumor from immune-mediated rejection [12,13].

PD-L1 expression in tumor cells is associated with poor disease outcomes in various cancers, including breast cancer [14,15,16,17,18,19,20]. PD-L1 expression in breast cancer has been correlated with positive lymph nodes, estrogen receptor (ER)-negativity, and TNBC [21,22]. A high level of PD-L1 in breast cancer is a predictor of poor overall survival [21,23]. PD-L1 targeting therapy has improved the prognosis of various cancers (e.g., melanoma, non-small-cell lung cancer, urothelial, renal cell, head and neck cancers, and lymphoma) [24]. It has recently been used for treating breast cancer, either as monotherapy or in combination with chemotherapy/neoadjuvant therapy, particularly in the TNBC, and it has demonstrated promising clinical outcomes [25,26,27,28,29]. However, this therapy’s benefit is currently limited to a small portion of patients, and the selection of patients for the treatment remains challenging [25,26]. PD-L1 tumor expression determined by immunohistochemistry (IHC) or the percentage of tumor-infiltrating lymphocytes (TILs) was not found to be associated with the response to anti-PD-L1 agents [30,31]. PD-L1 expression can change due to the source of the specimen. It could vary due to surgical resection, biopsy, primary tumors, metastasis tumors, prior treatment status, archival vs. fresh frozen tissues, and immune cell interactions [32,33].

Elevated serum PD-L1 has recently been identified as a poor prognostic factor in several cancer types, including multiple myeloma gastric cancer, thyroid gland carcinoma, non-small cell lung cancer (NSCLC), breast cancer, and soft tissue sarcomas [34,35,36,37,38,39,40]. Even serum PD-L1, also referred to as soluble PD-L1, to be considered a reliable treatment and disease progression measurement. However, PD-L1 serum/plasma’s clinical significance needs to be further understood, especially for breast cancer. The regulatory roles and function of PD-L1 in association with immune checkpoint blockade treatment are not fully understood.

The purpose of this study was to evaluate PD-L1 serum expression and its association with breast cancer, with focus on African-American and Latina women.

## 2. Materials and Methods

### 2.1. Human Subjects

After our University’s Institutional Review Board (# IRB 00-06-041) approval, the study population was recruited from the SPA6 region of South Los Angeles County in California. The population by race/ethnicity in SPA6 is 28% African-American, 68% Hispanic/Latino, and 4% other including Caucasian, Asian, Native-American, and Pacific-islander. The cohort comprised women examined in the Mammography Clinic or the Hematology/Oncology Clinic at the Martin Luther King Ambulatory Care Center (MACC, formerly known as King-Drew Medical Center) between 1998 and 2019. Women consented to an ongoing breast cancer study conducted in the Division of Cancer Research and Training at Charles R. Drew University of Medicine and Science and MACC. The Institutional Review Board approved the study. For follow-up data, we conducted post-hoc medical record abstraction. The inclusion/exclusion criteria were as follows. (a) Self-identified race/ethnicity: 30% were African-American, 65% were Hispanic/Latina, and the remaining 5% were Caucasian or Asian subjects. Considering that the number of Caucasian and Asian participants was relatively small and may not have generated meaningful statistical analysis, we only included African-American and Hispanic/Latina women in this study. (b) Breast cancer status was confirmed by the biopsy/pathology of the breast tissue, and only subjects who had documentation of this information were included in the study. Controls were participants with no diagnosis of cancer from mammograms and pathology results.

Benign cases were confirmed after an abnormal mammogram but with no malignancy from follow-up pathology results on the biopsy. (c) Baseline blood sample (serum sample collected at the time of diagnosis and before cancer treatment for cases). (d) Finally, we selected cases with documented disease follow-up information and controls, with the benign disease having 2-year follow-up mammography information for this study. The 394 subjects who fulfilled our inclusion criteria included 244 cases and 150 controls and were retrospectively selected for analysis.

### 2.2. Demographic and Clinical Information

Ethnicity was determined from self-reports at the time of diagnosis and recruitment; age, body mass index (BMI), and clinical data were obtained through medical chart extraction. Estrogen and progestogen receptor (ER/PR) status was considered “positive” if >1% of tumor cell nuclei were immunoreactive and “negative” if otherwise. HER2 (HER2/neu) status was considered” positive” if HER2 was 3+ and “negative” if HER2 was 0, 1+, or 2+, as determined by immunohistochemistry, or it was assigned using in situ hybridization to assess HER2 gene amplification. Tumor size, lymph node status, and TNM staging were all determined according to AJCC definitions. Tumor subtype was categorized according to the status of receptors as follows: (i) ER/PR+/HER2−; (ii) HER2+ (ER/PR− or ER/PR+); (iii) Triple-negative (ER/PR−/HER2−). Disease-free survival (DFS) was assessed based on screening tests such as mammography, CT scans, ultrasounds, bone scans that the patient underwent after treatment, and the primary cancer resolution. DFS for a patient was defined as not having any of the following: (1) reoccurrence, (2) metastatic disease, or (3) new primary tumor formation at another organ site.

### 2.3. Serum PD-L1 and Cytokine Levels

PD-L1 was measured using Human PD-L1 Quantikine ELISA Kit (Cat No. DB7H10) from the R&D Biosystem (Minneapolis, MN, USA) following the manufacturer’s recommendations. Briefly, 100 µL/well patient serum or plasma were incubated for 2 h at room temperature (RT) on an orbital shaker platform. After washing, 200 μL of Human/Cynomolgus Monkey B7-H1 Conjugate were added and incubated for 2 h on a shaker. Incubation was followed by adding 200 μL of substrate solution to each well for 30 min in the dark without shaking. Then, 50 μL of stop solution were added, and absorbance was measured at 540 nm. Each sample was analyzed in duplicate. Measured PD-L1 was derived from a standard curve generated each time the assay was performed. The levels of TNFα and IFNα2 were determined by Luminex multiplex assay on a Luminex 200 instrument (Luminex, Austin, TX, USA). The following Millipore Sigma (Billerica, MA, USA) kits were used: MILLIPLEX MAP TGF-β-3 Plex (Cat No. TGFBMAG-64K-03) and MILLIPLEX MAP Human Cytokine/Chemokine Panel 1 (Cat. No. HCYTOMAG-60K-15C) according to the manufacturer’s recommendations. Briefly, 25 µL of serum or plasma per well were incubated overnight with cytokine-specific magnetic beads at 4 °C. The following day, the samples were washed and incubated with 50 µL of detection beads at RT. Both incubations were performed on a plate-shaker at 800 rpm. Cytokine detection was performed on a Luminex 200 using 100 µL of xMAP™ Sheath Fluid (Cat no. 4050015, Thermo Fisher Scientific, Carlsbad, CA, USA). Data were analyzed using MILLIPLEX™ Analyst v5.1 (Virgene Tech, Carlisle, MA, USA). Each sample was analyzed in duplicate.

### 2.4. PTEN, pAkt, CD44, and CD24 Expression in Breast Cancer Tissue

The expression of PTEN, pAkt, WNT3, CD44, and CD24 in primary breast cancer tissue from the same cohort of breast cancer patients was evaluated by immunohistochemistry (IHC) in our previous study [3,41]. Briefly, paraffin blocks for tumor tissue containing more than 10% of tumor cells were selected for IHC. CD44/CD24 expression was determined by double-IHC staining. CD44 (Ab-4, NeoMarkers, Fremont, CA, USA; ready to use) was incubated for 60 min at RT and detected with Permanent Red (Vector Lab, Burlingame, CA, USA). CD24 (Ab-2 (SN3b), NeoMarkers, Fremont, CA, USA; 1:50 dilution) was incubated for 30 min at RT and detected using diaminobenzidine (DAB) (Vector Lab, Burlingame, CA, USA). Specific antibodies determined PTEN and phosphor-Akt (Ser473) (pAkt) expression against PTEN (Clone 6H2.1, DAKO, Santa Clara, CA, USA) and pAkt (#9271; Cell Signaling Technology, Inc., Danvers, MA, USA). Positive control paraffin slides with the known negative or positive expression of CD24, CD44, PTEN, and pAkt (IHC confirmed and antibody supplied by the vendor) was tested alongside the unknown samples. The proportion of CD44high/CD24low tumor cells was determined as the percentage of cells positive for Permanent Red staining but negative for DAB staining. The frequencies of CD44low/CD24high cells were determined similarly. The CD44 and CD24 double-staining procedure’s reliability was verified by single-staining with CD44 and CD24 antibodies. The evaluation of PTEN was exclusively based on positive cytoplasm staining, and PTEN-positive status was defined as >5% immunoreactive in tumor tissue. Each tissue was evaluated and scored by two clinical pathologists who were blinded to the tissue’s origin. Only those tissue sections that had good tissue structure and clear staining were included for analysis. A total of 81 cases were identified as having matched PD-L1 levels and were included for analysis in this study.

### 2.5. Statistical Analysis

Statistical analysis was performed using SPSS software (IBM SPSS Statistics version 22, IBM, Armonk, NY, USA). The normality of the distribution of PD-L1 was tested. The Shapiro-Wilk test showed a significant departure of serum level of PD-L1 from normality, W (233) = 0.89, *p* < 0.001. Hence, the level of PD-L1 was presented as the median level in this study. The statistical differences of median levels of PD-L1 among different ethnicities, ages (categorized as ten years group), BMI levels (classified as obese for BMI ≥ 30, overweight for BMI range 26–29, and normal for BMI ≤ 25; no subject had BMI < 18.5 in this cohort), and cancer characteristics were compared with Mann–Whitney U (2 samples) or Kruskal–Wallis one-way ANOVA (k samples) tests. Logistic regression with multivariate analysis was used to assess the association of deficiency in PD-L1 level with breast cancer in the total (adjusted for ethnicity, age, and BMI) and ethnically sub-categorized cohort (adjusted for age and BMI). According to their median levels, the levels of IFNα2 and TNFα cytokines were categorized as “high” or “low”. A median level of IFNα2 > 17 pg/mL was classified as “high”, and IFNα2 ≤ 17 pg/mL was categorized as “low”. A median level of TNFα > 15 pg/mL was categorized as “high,” and TNFα ≤ 15 pg/mL was classified as “low”. The statistical differences of PD-L1 between the high and low levels of IFNα2 and TNFα were assessed by the Mann–Whitney U test (two samples). The differences in the DFS of patients with different levels of PD-L1 were evaluated with Kaplan–Meier survival analysis with the log-rank test. Cox regression with multivariate analysis adjusted for ethnicity, histologic tumor grade, histologic subtype, size, lymph node status, ER/PR and HER2 status, tumor staging and subtypes, chemotherapies, and age at the time of diagnosis was used to assess relative risk (RR) for reducing DFS in patients with different levels of PD-L1. Throughout all analyses, only a *p* < 0.05 was considered statistically significant.

## 3. Results

### 3.1. Study Population

The PD-L1 serum level was measured in 244 women with breast cancer and 150 women who did not have cancer. The self-identified ethnic distribution of the participants is described in Table 1. A total of 150 (38.1%) were African-American, of whom 112 (75%) were breast cancer patients and 38 (25%) were women without breast cancer. A total of 224 were Hispanic, pf whom 132 (59%) were breast cancer patients and 112 (41%) were women without breast cancer. The age range of women in this study was 28–79 years. African-Americans’ median age was 53 years among the breast cancer patients and 50 years among the women without breast cancer. Hispanics’ median age was 49 years for women with breast cancer and 48 years for women without cancer. The majority of women in this cohort had a BMI ≥ 30, which is in the obese range (Table 1).

### 3.2. PD-L1 Serum Level in African-American and Hispanic Women with and without Breast Cancer

Breast cancer cases had significantly higher PD-L1 serum levels than those without cancer (controls). The median serum PD-L1 levels were 54.4 and 62 pg/mL in African-American and Hispanic cases, respectively, compared to about 37 pg/mL in the control cases (Table 2). A slightly higher serum PD-L1 level was observed in Hispanic breast cancer patients than in African-American breast cancer patients, but the difference was not statistically significant. The PD-L1 level was not associated with age or BMI in our study cohort (Table 2). The elevated serum PD-L1 level was significantly associated with breast cancer in African-American and Hispanic women. As shown in Table 2, the odds of breast cancer were found to be more than five-fold higher in women with a serum PD-L1 level ≥50 pg/mL than women with a PD-L1 level <50 pg/mL in both African American and Hispanic cohorts based on multivariate analysis adjusted for age and BMI.

### 3.3. Serum PD-L1 Level Is Higher in HER2+ and TNBC

The association of serum PD-L1 levels and breast cancer clinicopathological features is shown in Table 3. PD-L1 levels were significantly higher in women with estrogen and progesterone receptor-negative (ER/PR−) tumors and later-stage diseases than those with ER/PR+ or early-stage disease. The association was significant in the African-American cohort but not in the Hispanic cohort. The PD-L1 serum level of Hispanic women with ER/PR+ tumors were significantly higher than African-Americans with ER/PR+ tumors. Since the HER2+ tumor included ER+/HER2+ and ER−/HER2+, the data in Table 3 show no significant difference in PD-L1 levels between HER2+ and HER2− tumors. However, compared to ER/PR+/HER2− tumors, the PD-L1 serum level was considerably elevated in HER2+ and TNBC subtype tumors (Table 3). Similar to the association with ER/PR status, the elevated PD-L1 serum level observed in African-American women was statistically significant (*p* = 0.03, tested with Kruskal–Wallis one-way ANOVA), but not in Hispanic women (*p* = 0.06, Kruskal–Wallis one-way ANOVA, Table 3). We found that the expression of PD-L1 was not associated with lymph node involvement or primary tumor size.

### 3.4. Increased PD-L1 in Breast Cancer Is Associated with High Levels of IFNα2 and TNFα

Interferons (IFNs) are a group of cytokines produced by various cells in the inflammatory response to infections. We examined the levels of these cytokines in the same cohort of patient sera in this study. The serum level of PD-L1 was found to be significantly correlated with the serum levels of IFNα2 (interferon-α, r = 0.2, *p* = 0.037) and TNFα (tumor necrosis factor-α, r = 0.3, *p* = 0.001). We also categorized IFNα2 and TNFα levels as “high” or “low” according to the median level, as described in the Methods section. Both IFNα2 and TNFα levels were significantly high in breast cancer patients with high PD-L1 levels (Figure 1A). The level of IFNγ was higher in patients with a high level of PD-L1; however, it was not significant (data are not shown). We did not see increased IFNα and TNFα levels in women with high PD-L1 levels in non-cancer controls.

### 3.5. A High Level of PD-L1 Is Associated with PTEN Loss and Cancer Stem-like Phenotype

We next tested whether PD-L1 levels were associated with the activation of the PI3K/Akt pathway and the presence of cancer stem cell-like cells in this cohort of patients. We compared the PTEN, pAkt, and CD44/CD24 protein levels in breast cancer tissue determined by IHC with the serum levels of PD-L1. The data in Figure 1B demonstrate that an increased serum level of PD-L1 was more likely to be in patients whose tumor expressing the CD44high/CD24low phenotype and with tumors presenting the loss of PTEN and high pAkt expression. Multivariate analysis further indicated that high serum PD-L1 is independently associated with the CD44high/CD24low phenotype (Table 4).

### 3.6. PD-L1 Serum Level Predicts Disease-Free Survival in Breast Cancer

We then examined the predictive role of serum PD-L1 level (the blood samples were collected prior to any treatment) for disease-free survival (DFS) in different subtypes of breast patients without metastatic diseases. We analyzed the association between serum PD-L1 levels and DFS in breast cancer patients. Kaplan–Meier survival analyses were conducted according to different levels of PD-L1 in our total cohort and specific cohorts of African-American and Hispanic participants. As shown in Figure 2, higher PD-L1 levels were found to significantly reduce 5-year DFS in the entire cohort and the African-American and Hispanic cohorts.

To assess the relative risk (RR) for reducing DFS, we performed Cox regression with multivariate analysis adjusted for ethnicity, age at the time of diagnosis, histologic tumor grade, histologic subtypes, tumor size, lymph node status, ER/PR and HER2 status, tumor staging, subtypes, and chemotherapies. Patients in the study were all treated at the Martin Luther King Ambulatory Care Center. Most of the patients received either CAF (cyclophosphamide, doxorubicin hydrochloride (Adriamycin), and 5-fluorouracil) or CA (cyclophosphamide, Adriamycin) and Taxotere. Some of the patients diagnosed in the early years received CMF (cyclophosphamide, methotrexate, and 5-fluorouracil). The data in Table 5 show that patients who had PD-L1 levels between 58 and 79 pg/mL had a 3.8-fold risk of cancer relapse or metastases compared to patients with PD-L1 levels of less than 58 pg/mL (*p* = 0.003). Similarly, for patients with PD-L1 levels >79 pg/mL, the RR was 3.6 (*p* = 0.043). When the analysis was performed by stratifying for ethnicity, we found that the RR was 1.1 in African-American patients with PD-L1 levels at the 58–79 pg/mL range, but it was not statistically significant. The RR was increased to 4.4 (*p* = 0.036) for African-American patients when the serum PD-L1 level was >79 pg/mL (Table 5). We found a significantly higher RR for reducing DFS with higher PD-L1 levels in the Hispanic cohort. As shown in Table 5, the RRs were 4.3 (*p* = 0.001) and 3.6 (*p* = 0.008) in Hispanic patients with PD-L1 levels at 58–79 and >79 pg/mL, respectively. The data indicate that serum PD-L1 levels can independently predict poor DFS for breast cancer patients. Even though the expression of PD-L1 in serum could be an essential biomarker for selecting breast cancer patients with immunotherapy, it must be compared with the current standard of care for determining patients for atezolizumab treatment and be established from a clinical trial.

## 4. Discussion

High PD-L1 expression in breast cancer has been associated with large tumor size and high-grade tumors with negative estrogen receptors and positive HER2 receptors [21,22], and it is linked to poor disease outcomes [21,23]. Findings from a meta-analysis included five studies containing 2546 breast cases indicating that PD-L1 expression in tumor tissues could be a valuable biomarker for breast cancer prognosis and patient selection for immunotherapy [23]. Recent therapies targeting PD-1/PD-L1 have been used in treating metastatic breast cancer and have shown promising activity [25]. However, various treatment responses have been ascribed to challenges in standardizing PD-L1 IHC test methods [30]. Furthermore, the immune checkpoint blockade mediators play a role in tumor progression. Still, they have been shown to be regulated by the tumor microenvironment [42,43,44], confounding interpretation of the expression of PD-L1 in tumor tissue. Hence, we tested whether PD-L1 levels in breast cancer patients’ serum may also be used as a biomarker in this study.

We compared serum PD-L1 levels in women with and without breast cancer. Our study comprised African-American and Hispanic women, who are more likely to express greater incidence of TNBC, poor DFS, and higher mortality than Caucasian women. Our data showed that PD-L1 serum levels were significantly higher in women with breast cancer than in normal controls. The elevated PD-L1 serum level was significantly associated with ER/PR− tumor and later-stage disease. Women, especially African-American women with the TNBC and HER2+ subtypes of breast cancer, had considerably higher levels of PD-L1 than women with the ER/PR+/HER2+ type of cancer. This study’s results were consistent with the data from the meta-analysis in which PD-L1 expression was tested in breast cancer tissues [23]. In the meta-analysis, Zhang et al. reported that PD-L1 expression in breast cancer tissues was associated with positive lymph node metastasis, higher histological grades, ER-negativity, and triple-negative breast cancer [23]. However, we did not find the association between PD-L1 serum levels with lymph node positivity in our current patients’ cohort. We also found significantly higher PD-L1 serum levels in Hispanic women with ER/PR+ tumors.

It has been reported that PD-L1 can deliver PI3K/Akt signals to tumor cells through PD-1 binding and lead to resistance to cancer treatment in multiple myeloma [45], as well as potentially enriching its expression in cancer stem-like cells (CSCs) [46]. Our previous studies evaluated PTEN and pAkt protein and CD44/CD24 CSC marker expression in breast cancer tissues in the same cohort of African-American and Hispanic patients [3,41]. Data from those studies showed that patients’ TNBC cohort had a significant association with PTEN loss and the CD44high/CD24low phenotype [3]. In addition, the tumor tissues with PTEN loss had increased pAkt expression. The increased pAkt expression was also more frequent in patients with HER2+ tumors in this cohort of women [41]. In this study, we observed that a high PD-L1 serum level was significantly associated with tumor tissue with high pAkt and PTEN loss in the same cohort. In addition, the tumor cells exhibiting CD44high/CD24low presented significantly elevated PD-L1 serum levels. We performed multivariate analysis adjusted for breast cancer subtypes to verify the significance of the association of PD-L1 levels with pAkt, PTEN, and CD44high/CD24low. Our data indicated that the CD44high/CD24low phenotype was independently associated with the level of PD-L1 (OR = 7.0, *p* = 0.01). CD44 has been reported to be positively correlated with PD-L1 expression at the mRNA and protein levels in primary tumor samples of TNBC and non-small cell lung cancer (NSCLC) patients [47]; that study identified the cell-surface adhesion receptor CD44 as a critical positive regulator of PD-L1 expression in these cancers, and CD44 mechanistically activates PD-L1 transcription in part through its cleaved intracytoplasmic domain [45]. Hsu et al. found that epithelial–mesenchymal transition (EMT) enriches PD-L1 in CD44high/CD24low cells through the EMT/β-catenin/STAT3/PD-L1 signaling axis, and the enriched PD-L1 expression in CD44high/CD24low contributes to CSCs’ immune evasion [46]. CD44, a well-established CSC marker, is necessary to initiate EMT and is linked to cancer treatment resistance and poor outcome [44]. The high PD-L1 expression in CD44high cells in TNBC may lead to advanced tumorigenic, immunosuppressive, and chemo-resistant features, as well as contribute to tumor metastasis and progression. Hence, the CD44/PD-L1 axis could be a critical therapeutic target for treating TNBC and other breast cancer subtypes. Moreover, PTEN loss also has been reported as one of the mechanisms regulating PD-L1 expression [22]. Mittendorf et al. demonstrated that the inhibition of the PI3K pathway with the AKT inhibitor MK-2206 or rapamycin decreased PD-L1 expression and provided evidence linking PTEN and PI3K signaling to PD-L1 [22].

Cytokines in the tumor microenvironment may upregulate the PD-L1 level. It has been reported that cytokines, IFNs, and TNFα regulate the inducible expression of PD-L1 on breast cancer and immune cells [42,43,48,49]. IFNγ and its signaling induce PD-L1 in TNBC [48,49], IFNα upregulates PD-L1 in dendritic cells [42], and TNFα upregulates PD-L1 in prostate and colon cancer cells [50]. This study revealed a significant association between serum PD-L1 and IFNα2 and TNFα levels in breast cancer, though not in normal controls. Mechanistic studies are required to understand how elevated PD-L1 serum levels are linked to increased cytokines.

Furthermore, the data from the meta-analysis showed that the combined hazard ratio (HR) for overall survival was 1.76 (*p* = 0.02) for patients with tumors overexpressing PD-L1. Therefore, PD-L1 expression could be a promising biomarker for breast cancer prognosis [23]. Muenst et al. tested the PD-L1 levels of 650 breast cancer patients with IHC analysis and found that PD-L1 expression is an independent negative prognostic factor in human breast cancer [21]. However, the PD-L1 level in these studies was only assessed in breast cancer tissues with IHC. Compared to others, our study’s novelty was that we evaluated serum PD-L1 in the most underserved and underrepresented communities. Predicting makers for breast cancer progression will significantly impact breast cancer assessment, treatment, and care.

High baseline serum PD-1 or PD-L1 levels have been shown to predict poor PD-1 inhibition therapy outcomes in metastatic melanoma [33,34], indicating that the serum PD-L1 level may be used as a predictive marker assessing effects of anti-PD-1/PD-L1 therapies. Data from this study show the first time that DFS was significantly reduced for African American and Hispanic/Latina women with high PD-L1 serum levels. Multivariate analysis adjusted for tumor histologic grade, histologic subtypes, size, lymph node status, ER/PR and HER2 status, tumor subtypes, staging, and age indicated that elevated PD-L1 serum level independently predicts poor DFS in African-American and Hispanic women. The data suggest that the PD-L1 serum level at the time of diagnosis can serve as a predictive marker for cancer outcomes and potentially be used for patient selection for anti-PD-L1 therapy.

The current standard of care protocol recommends selecting patients for the immunotherapy for metastatic TNBC (mTNBC) patients based on their PD-L1 expression greater than or equal to 1% in fixed tumor tissue samples. This includes PD-L1-stained tumor-infiltrating immune cells of any intensity covering ≥1% of the tumor area [51]. Data from the IMpassion 130 phase III clinical trial showed that atezolizumab plus nab-paclitaxel prolonged progression-free survival among patients with mTNBC in both the intention-to-treat population and the PD-L1-positive subgroup [51]. However, treatment with atezolizumab and paclitaxel did not significantly reduce the risk of cancer progression and death compared to placebo and paclitaxel in the PD-L1-positive population trial [51,52]. The majority of participants were Caucasians, with fewer African Americans (intention-to-treat population =59 and PD-L1-positive subgroup =23) [51].

We acknowledge that we have not correlated PD-L1 expression levels in tumor tissues by IHC with circulating or secreted PD-L1 levels in the serum samples from the same patient in our current study. To establish the serum PD-L1 level as a biomarker, further validation studies are needed to compare the benefit of using laboratory-based tumor pathology with the serum levels of PD-L1 to select TNBC and other breast cancer patients for immune checkpoint inhibitor treatment options. Additionally, further studies are warranted to determine whether the serum level of PD-L1 could serve as a biomarker for patient selection for anti-PD-L1 therapy and assess treatment outcomes. We will continually verify the consistency and differences between serum and tissue levels in further studies. In addition, the identity of African Americans and Latinx was based on self-identify for this study. We could not more accurately identify the races/ethnicity since both African Americans and Latinx are mixed-race. Testing ancestry makers to identify the race and ethnic group from those patients can be carried out in our laboratory, which will allow us to have accurate interpretations.

Nonetheless, our study provides evidence that serum PD-L1 in breast cancer before treatment can predict DFS. Thus far, there has been a dearth of information on assessing PD-L1 levels in African Americans and Latinx, who are frequently diagnosed at advanced stages and have poor clinical outcomes.

The study’s innovative points are: (1) the serum level of PD-L1 could be a convenient and critical biomarker for selecting patients for immunotherapy, and it may also serve as a real-time biomarker for monitoring therapeutic efficacy; (2) the study has shown that African American women with TNBC and a high baseline level of serum PD-L1 may benefit from anti-PD-L1 therapies; (3) we demonstrated an interesting clinical correction between serum PD-L1 levels and cytokines, as well as association with cancer stem cell markers and other oncogenic markers from the corresponding patients’ cancer tissues.

## 5. Conclusions

This study suggests that in African-American and Hispanic breast cancer patients, serum PD-L1 level may predict DFS. Further validation studies are needed to compare the benefits of using laboratory-based tumor pathology with the serum levels of PD-L1 to select TNBC and other breast cancer patients for immune checkpoint inhibitor treatment options.

## Figures and Tables

**Figure 1 jcm-11-00283-f001:**
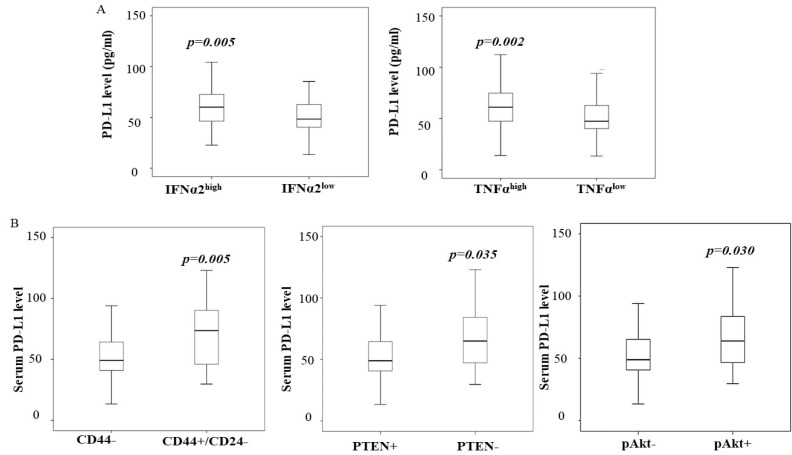
High serum PD-L1 level is associated with an aggressive phenotype. (**A**) Serum levels of IFNα2 and TNFα were measured by Luminex multiplex assay, following the manufacture’s instruction described in the Methods section, and categorized as “high” and “Low” (as described in the Methods section). Comparison of serum PD-L1 level with IFNα2 and TNFα levels in breast cancer; IFNα2 high vs. IFNα2 low (*p* = 0.005, left panel) and TNFα high vs. TNFα low (*p* = 0.002, right panel); (**B**) comparison of PD-L1 serum level with PTEN phosphorylated Akt (pAkt) and CD44/CD24 protein levels in breast cancer tissue. The box graphs indicate the serum PD-L1 level vs. levels of PTEN, pAkt, and CD44/CD24. CD44high/CD24low vs. CD44 low, *p* = 0.005; PTEN− vs. PTEN+, *p* = 0.035 and pAkthigh vs. pAktlow, *p* = 0.03. Each box graph shows the median and the range.

**Figure 2 jcm-11-00283-f002:**
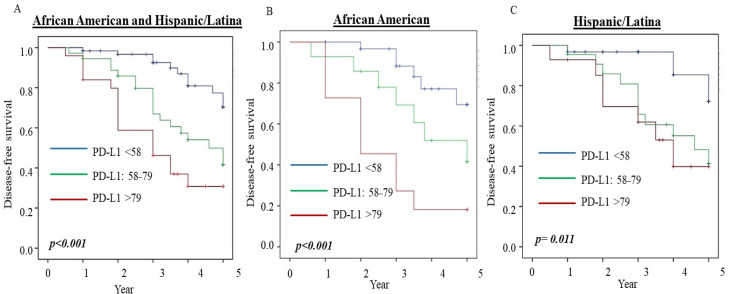
Five-year disease-free survival and PD-L1 level. PD-L1 levels were categorized as different levels according to percentiles, as estimated with Kaplan–Meier Survival analysis. The log-rank test determined significance. (**A**) DFS by PD-L1 levels in African-American and Hispanic/Latina patients (N = 239, PD-L1 < 58 pg/ml = 117, PD-L1: 58–79 pg/mL = 61, PD-L1 > 79 pg/mL = 61); (**B**) DFS by PD-L1 levels in African-American patients (N = 112, PD-L1 < 58 pg/mL = 61, PD-L1: 58–79 pg/mL = 25, PD-L1 > 79 pg/mL = 26); (**C**) DFS by PD-L1 levels in Hispanic/Latina patients (N = 127, PD-L1 < 58 pg/mL = 56, PD-L1: 58–79 pg/mL = 36, PD-L1 > 79 pg/mL = 35).

**Table 1 jcm-11-00283-t001:** Study population.

	African-American (n = 150)			Hispanic (n = 244)		
	Cases (n = 112) Median (Range)	Controls (n = 38) Media (Range)	*p*	Cases (n = 132) Median (Range)	Controls (n = 112) Media (Range)	*p*
Age (years)	^ 53 (32–79)	^^ 50 (27–70)	0.33	^ 49 (28–77)	^^ 48 (22–77)	0.27
*Percentiles*						
25	46	46		42	40	
50	53	50		49	48	
75	57	57		56	54	
BMI ^ψ^	31 (22–55)	32 (21–52)	0.97	31 (20–57)	30 (20–53)	0.34
*Percentiles*						
25	28	29		27	27	
50	31	31		31	30	
75	38	38		35	35	
	N (%)	N (%)		N (%)	N (%)	
Obesity *	65 (63.1)	21 (63.6)		66 (54.5)	51 (50.0)	
Overweight **	23 (22.3)	9 (27.3)		38 (31.4)	37 (36.3)	
Normal (BMI ≤ 25)	15 (14.6)	3 (9.1)	0.68	17 (14.0)	14 (13.7)	0.66

* BMI ≥ 30; ** BMI <30 and >25; ^^ *p* = 0.003; ^ *p* = 0.05; ^ψ^ Body Mass Index. *p*-values were tested with the Mann–Whitney U test for African American vs. Hispanic and wit the Kruskal–Wallis one-way ANOVA test for obesity vs. overweight vs. normal.

**Table 2 jcm-11-00283-t002:** Serum PD-L1 level in African American and Hispanic/Latina women with and without breast cancer.

	African American (n = 150)			Hispanic (n = 244)		
	Cases (n = 112) Median (Range)	Controls (n = 38) Median (Range)	*p*	Cases (n = 132) Median (Range)	Controls (n = 112) Median (Range)	*p*
PD-L1 (pg/mL)	54.4 (20.6–206.1)	37.3 (16–78.3)	<0.001	62.0 (13.4–221.7)	37.1 (13.2–87.1)	<0.001
*Percentiles*						
25	41.2	32.2		43.5	29.7	
50	54.4	37.3		61.9	37.1	
75	76.6	44.9		81.5	50.4	
Age (years)	N Median (range)	N Median (range)		N Median (range)	N Median (range)	
≤30	- -	1 42.6		2 37.9 (26.6–49.2)	55 4.6 (37.2–59.2)	0.190
31–40	8 51.7 (23.8–182.2)	4 31.2 (22.9–42.4)	0.11	25 57.9 (25.3–103.1)	23 33.7 (13.2–81.8)	<0.001
41–50	37 53.8 (20.6–88.6)	15 37.0 (16.0–69.8)	0.01	47 58.7 (13.4–116.7)	39 33.2 (16.2–78.1)	<0.001
51–60	49 48.6 (21.3–117.4)	14 38.4 (26.6–78.3)	0.05	40 64.5 (13.9–221.7)	34 40.5 (18.1–87.1)	<0.001
≥60	18 62.6 (27.5–206.1)	4 37.7 (32.3–44.3)	0.007	18 65.4 (37.6–161.0)	11 48.1 (27.2–77.5)	<0.001
	N Median (range)	N Median (range)		N Median (range)	N Median (range)	
Obesity *	65 56.0 (21.3–182.2)	21 37.3 (25.1–78.3)	<0.001	66 64.6 (13.9–116.7)	51 36.3 (13.2–87.1)	<0.001
Overweight **	23 53.4 (24.3–206.1)	9 34.5 (16.0–51.9)	0.006	38 58.8 (13.4–131.4)	37 37.1 (18.1–77.5)	<0.001
Normal	15 58.7 (23.8–117.4)	3 44.0 (42.6–61.3)	0.36	17 52.0 (26.6–161.0)	14 38.0 (24.3–1.1)	0.04
	Cases vs. Control			Cases vs. Control		
	^OR	95% CI	*p*	^ OR	95% CI	*p*
PD-L1 (pg/mL)						
≤50	1			1		
>50	5.1	1.9–13.7	0.001	5.3	2.9–9.6	<0.001

* BMI ≥ 30; ** BMI <30 and >25, ^ Odds ratio. The Mann–Whitney U test was used to measure *p*.

**Table 3 jcm-11-00283-t003:** Serum PD-L1 level and breast cancer.

	Total	African-Americans	Hispanics	
	N PD-L1 (pg/mL) Median (Range)	N PD-L1 (pg/mL) Median (Range)	N PD-L1 (pg/mL) Median (Range)	* *p*-Value
**Cases** **Controls** ***p*-Value**	244 58.3 (13.4–221.7)	112 54.4 (20.6–206)	132 61.9 (13.4–221.7)	*0.12*
	150 37.2 (13.2–81.7)	38 37.3 (16–78.3)	112 37.1 (13.2–87.1)	*0.82*
	** *<0.001* **	** *<0.001* **	** *<0.001* **	
**ER/PR**				
**Positive**	150 50.1 (13.4–221.7)	64 47.3 (21.3–206.1)	86 58.7 (13.4–221.7)	** *0.03* **
**Negative**	94 64.3 (20.6–182.2)	48 63.1 (20.6–182.2)	46 66.4 (25.3–161.0)	*0.71*
*** *p*-Value**	** *0.005* **	** *0.005* **	*0.17*	
**HER2**				
**Positive**	52 64.7 (20.6–161.0)	20 62.7 (20.6–103.0)	32 64.8 (25.3–161.1)	*0.21*
**Negative**	192 56.0 (13.4–221.7)	89 53.9 (21.3–206.1)	103 58.7 (13.4–221.7)	*0.36*
*** *p*-Value**	*0.35*	*0.92*	*0.25*	
**Tumor size**				
**T0-T1**	64 58.7 (21.3–206.1)	35 50.5 (21.3–206.1)	29 60.3 (31.8–131.4)	*0.27*
**T2**	117 56.0 (13.9–221.7)	52 53.9 (24.3–106.0)	65 61.8 (13.9–221.7)	*0.25*
**T3-T4**	63 63.9 (13.4–182.2)	25 63.4 (20.6–182.0)	38 64.3 (13.4–139.0)	*0.87*
**** *p*-Value**	*0.47*	*0.32*	*0.88*	
**Lymph node**				
**Negative**	111 56.0 (13.9–206.1)	51 54.9 (20.6–206.1)	61 58.7 (13.9–139.0)	*0.31*
**Positive**	122 62.2 (13.4–221.7)	54 56.6 (24.3–182.2)	67 64.7 (13.4–221.7)	*0.43*
*** *p*-Value**	*0.32*	*0.77*	*0.29*	
**TNM staging**				
**I/II**	168 54.1 (13.9–206.1)	84 50.0 (20.6–206.1)	84 58.8 (13.9–131.4)	*0.09*
**III/IV**	76 66.6 (13.4–221.7)	28 69.0 (31.0–182.2)	48 65.1 (13.4–221.7)	*0.50*
*** *p*-Value**	** *0.01* **	** *0.04* **	*0.20*	
**Subtype**				
**ER/PR+/HER2−**	128 49.1 (13.4–221.7)	57 47.3 (21.3–206.1)	71 52.6 (13.4–221.7)	*0.17*
**HER2+**	52 64.6 (20.6–161.0)	20 62.7 (20.6–103.0)	32 64.8 (25.3–161.0)	*0.26*
**ER/PR−/HER2−**	64 64.0 (26.6–182.2)	35 61.6 (26.6–182.2)	29 66.9 (35.2–123.0)	*0.48*
**** *p*-Value**	** *0.003* **	** *0.03* **	** *0.064* **	

* *p* was tested with the Mann–Whitney U test (2 samples); ** *p* was tested with the Kruskal–Wallis one-way ANOVA (k samples) test.

**Table 4 jcm-11-00283-t004:** Multivariate analysis: PD-L1 serum level, tumor subtypes, and protein markers.

	PD-L1 ≥ 79 pg/mL vs. PD-L1 < 79 pg/mL		
	OR	95% CI	*p*
**Subtypes**			
*ER/PR+/HER2−*	1		
*HER2+*	0.7	0.3–3.6	0.63
*TNBC*	1.4	0.3–7.4	0.71
**pAkt ***			
*Low*	1		
*High*	0.6	0.3–4.4	0.71
**PTEN**			
*Positive*	1		
*Negative*	2.9	0.6–15.4	0.65
**CD44high/CD24low**			
*Negative*	1		
*Positive*	7.0	1.5–32.1	**0.01**

* Phosphorylated Akt. *p* was obtained by logistic regression with multivariate analysis.

**Table 5 jcm-11-00283-t005:** The relative risk of reducing disease-free survival by PD-L1 level.

PDL-1 (pg/mL)	Multivariate Analysis		
	RR	95% CI	*p*-Value
**Total ^**			
PD-L1 < 58	1		
PD-L1: 58–79	3.8	1.9–13.9	** *0.003* **
PD-L1 > 79	3.6	1.5–15.3	** *0.043* **
**African-American ^^**			
PD-L1 < 58	1		
PD-L1: 58–79	1.1	0.2–5.5	*0.927*
PD-L1 > 79	4.4	1.1–36.4	** *0.036* **
**Hispanic/Latina ^^**			
PD-L1 < 58	1		
PD-L1: 58–79	4.3	1.6–23.6	** *0.001* **
PD-L1 > 79	3.6	1.3–24.9	** *0.008* **

^ Adjusted for tumor histologic grade, histologic subtypes, size, lymph node status, ER/PR and HER2 status, tumor subtypes, stages, treatment, age at the time of diagnosis, and ethnicity, ^^ Adjusted for tumor histologic grade, histologic subtypes, size, lymph node status, ER/PR and HER2 status tumor subtypes, stages, treatment, and age. *p* was obtained by Cox regression with multivariate analysis. Italics and Bolded numbers imply significant changes.

## Data Availability

The data presented in this study are available on request from the corresponding author and principal investigator, though they restricted to investigators based in academic institutions.

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
