# Peer review of "Elevated Baseline Serum PD-L1 Level May Predict Poor Outcomes from Breast Cancer in African-American and Hispanic Women"

_jcm, 2022, doi:10.3390/jcm11020283_

Round 1

Reviewer 1 Report

The manuscript provides novel data confirming known association of PD-L1 expression with breast cancer outcomes in African-American and Hispanic women. It is described well and is scientifically sound. The authors may want to address the following minor comments:

Fig.1: for the data like this, why was not correlation analysis done, instead of dividing groups into high/low markers expression and comparing them?

Lines 444-447: This study by Muenst and some others cited: this manuscript results seem to repeat whatever results in the papers cited. What is the difference/novelty of what the authors have done? (I would guess, the ethnicity of the patients cohort?) Please emphasize it, here and in some other placed throughout the Discussion.

Lines 481-483: No correlation was demonstrated but comparisons between high and low markers expression groups.

Lines: 479-480: There are no evidence or results obtained in this manuscript that would support this. PD-L1 was not studied here as a marker specifically for monitoring therapeutic efficacy. There are papers cited in the manuscript that show it, but then it cannot be presented as a conclusion of this manuscript.

Author Response

We wish to thank Reviewer #1 for the helpful and constructive suggestions.

Please see our detailed responses below.

Response to reviewer's comments

Reviewer #1

The manuscript provides novel data confirming the known association of PD-L1 expression with breast cancer outcomes in African-American and Hispanic women. It is described well and is scientifically sound. The authors may want to address the following minor comments:

Response: Dear Reviewer 1. Thank you for your overall very positive comments and encouragement for our study. We have responded accordingly to each of your concerns and recommendations.

Fig.1: for the data like this, why was not correlation analysis done instead of dividing groups into high/low markers expressions and comparing them?

Response: According to the Reviewer's suggestion, we have conducted a correlation analysis between PD-L1 and IFNα and PD-L1 and TNFα Reviewer's suggestion. The serum level of PD-L1 was significantly correlated with serum levels of IFNα2 (r=0.2, p=0.037) and TNFα (r=0.3, p=0.001). The data have been added in the revised version of the manuscript in Lines: 273-275.

As we mentioned in figure 1 legend and method section, serum levels of IFNα2 and TNFα were measured by the Luminex multiplex assay. PTEN phosphorylated Akt (pAkt) and CD44/CD24 protein levels in breast cancer tissues were determined by IHC. Hence, the comparison was made by dividing the groups into high/low for marker(s) expression and comparing them. We also edited the text in lines 275-281 and 293-297 to explain the data in Figure 1 in the revised manuscript. 

Lines 444-447: This study by Muenst and some others cited: this manuscript results seem to repeat whatever results in the papers cited. What is the difference/novelty of what the authors have done? (I would guess, the ethnicity of the patient's cohort?) Please emphasize it here and in some other places throughout the Discussion.

Response: Thank you for the Reviewer's suggestions. We have revised the Discussion section and emphasized that our study focused on African American and Hispanic/Latinx women with breast cancer. It is critical to emphasize that our study focuses on populations most impacted by cancer health disparities. Hence, our study's novelty compared to others is evaluating serum PD-L1 in the most underserved and underrepresented communities. Predicting makers for breast cancer progression will significantly impact breast cancer assessment, treatment, and care. Please see the revised version (lines: 435-438 and 442-443).

Lines 481-483: No correlation was demonstrated, but comparisons between high and low markers expression groups.

Response: In the revised version, we have added results from correlation analysis between PD-L1 and cytokines, TNFα and IFNα2 (line: 273-275). We also edited the text in Discussion (lines: 488-491)

Lines: 479-480: No evidence or results obtained in this manuscript would support this. PD-L1 was not studied here as a marker specifically for monitoring therapeutic efficacy. The papers cited in the manuscript show it, but it cannot be presented as a conclusion.

Response: Thank you. We agree. We have edited the Discussion and Conclusion sections and deleted the sentence "serum PD-L1 level could be a potential biomarker for selecting patients for anti-PD-L1 therapy" in the revised version. The conclusion in the abstract is also revised.

Reviewer 2 Report

The manuscript submitted by Wu et al. investigated the clinical significance of measuring serum PD-L1 levels in African-American and Hispanic women diagnosed with breast cancer in relation to a cohort of patients without breast cancer. In addition, the correlation between PD-L1 serum level and IFNα2, TNFα serum levels was analyzed.  Also, breast cancer tissue PTEN, Akt, and CD44/CD24 protein levels expression in immunohistochemistry was evaluated by comparison with PD-L1 serum level. To this end, the authors conducted a retrospective observational study, comprising 244 patients which presented with TNBC (ER/PR-/HER2-), HER2 positive (HER2+/ER/PR-) and HER2 negative (ER/PR+/HER2-) breast cancer between 1998 and 2019 in two institutions, which were compared with 150 patients without breast cancer, in terms of clinical and demographical data, pathological data, laboratory data (PD-L1 levels, TNFα levels, IFNα2), as well as disease-free survival. The authors concluded that in African-American and Hispanic breast cancer  patients serum PD-L1 level may predict disease-free survival and could be a potential biomarker for selecting patients for anti-PD-L1 therapy. The main strength of this study is that it addresses a relevant research question, with significant implications for clinical practice. This study could facilitate novel approaches in targeted therapy for breast cancer including molecular forms that are resistant to standard oncologic treatment regimens.

Title and abstract: The title and abstract are appropriate for the contents of the text.

Introduction: The authors summarized the current available information on this topic in a clear and concise manner. The final  paragraph in this section is somehow inappropriate due to the fact that the authors present the results and conclusions of the present study. Perhaps this information should be discussed in the discussion section.

Materials and methods: The patients appear to represent the whole experience of the investigators. The methodology for patient inclusion and exclusion was presented clearly.  The methodology for determining cytokines levels and PD-L1 levels adequately explained. The statistical analysis is described, however, it is unclear whether the normality of the distribution of data was tested (for example, if a Shapiro-Wilk test was performed).

Results: The authors adequately presented their findings. The information presented is nicely supported by the figure and tables. However the authors reference results and explanations from previously published papers (e.g. lines 270, 272, 292, 297, 304, etc.).  This instances should be revised and the references should be included in the discussion section.

Discussions: The results are discussed in relation to the evidence currently available in the literature. The limitations and strengths of the present study are adequately presented.

Conclusions: The conclusions of the authors are appropriately cautious given the limitations of the study.

Lastly, the use of language is sound and no revision of grammar and syntax is required.

Author Response

We wish to thank Reviewer #2 for the helpful and constructive suggestions.

Please see our detailed responses below.

 Response to reviewer's comments

Reviewer #2

The manuscript submitted by Wu et al. investigated the clinical significance of measuring serum PD-L1 levels in African-American and Hispanic women diagnosed with breast cancer in relation to a cohort of patients without breast cancer. In addition, the correlation between PD-L1 serum level and IFNα2, TNFα serum levels were analyzed.  The authors concluded that in African-American and Hispanic breast cancer patients, serum PD-L1 level may predict disease-free survival and could be a potential biomarker for selecting patients for anti-PD-L1 therapy. Also, breast cancer tissue PTEN, Akt, and CD44/CD24 protein levels expression in immunohistochemistry was evaluated by comparison with PD-L1 serum level. To this end, the authors conducted a retrospective observational study, comprising 244 patients who presented with TNBC (ER/PR-/HER2-), HER2 positive (HER2+/ER/PR-), and HER2 negative (ER/PR+/HER2-) breast cancer between 1998 and 2019 in two institutions, which were compared with 150 patients without breast cancer, in terms of clinical and demographical data, pathological data, laboratory data (PD-L1 levels, TNFα levels, IFNα2), as well as disease-free survival. The main strength of this study is that it addresses a relevant research question with significant implications for clinical practice. This study could facilitate novel approaches in targeted therapy for breast cancer, including molecular forms that are resistant to standard oncologic treatment regimens.

Response: We also wish to thank Reviewer 2 for a very positive and encouraging assessment. We are grateful for the constructive recommendations that will strengthen the significance of our data and the publication.

Title and abstract: The title and abstract are appropriate for the contents of the text.

Introduction: The authors summarized the currently available information on this topic in a clear and concise manner. The final paragraph in this section is somehow inappropriate due to the fact that the authors present the results and conclusions of the present study. Perhaps this information should be discussed in the discussion section.

Response: Per the Reviewer's suggestion, we have edited the final paragraph in the introduction. Please see the revised version in lines: 87-89.

Materials and methods: The patients appear to represent the whole experience of the investigators. The methodology for patient inclusion and exclusion was presented clearly.  The methodology for determining cytokines levels and PD-L1 levels was adequately explained. The statistical analysis is described. However, it is unclear whether the normality of the distribution of data was tested (for example, if a Shapiro-Wilk test was performed).

Response: Yes, the normality of the distribution of PD-L1 was tested first before other data analyses. Shapiro-Wilk test showed a significant departure of serum level of PD-L1 from normality, W (233) = 0.89, P<0.001. Hence, the PD-L1 level was presented as median rather than mean level.

We have edited methods and indicated that the normality of the distribution of serum PD-L1 level was tested by the Shapiro-Wilk test (Lines: 190-192).

Results: The authors adequately presented their findings. The information presented is nicely supported by the figure and tables. However, the authors reference results and explanations from previously published papers (e.g., lines 270, 272, 292, 297, 304, etc.).  These instances should be revised, and the discussion section should include the references.

Response: We have removed the discussion about previous studies in the results section and integrated the relevant references and discussion into the revised Discussion section in the revised manuscript.

Discussions: The results are discussed in relation to the evidence currently available in the literature. The limitations and strengths of the present study are adequately presented.

Response: Once again, we are very grateful to the Reviewer for constructive comments

Conclusions: The conclusions of the authors are appropriately cautious given the limitations of the study.

Lastly, the use of language is sound, and no revision of grammar and syntax is required.

Response: We thank the Reviewer's their positive and constructive comments

Round 2

Reviewer 2 Report

I have examined the revised version of the manuscript submitted by Wu et al. titled “ Elevated baseline serum PD-L1 level may predict poor outcomes from breast cancer in African-American and Hispanic women”.

The authors managed to adequately address the previously raised issues concerning this paper, as follows:

Introduction: the final paragraph of this chapter has been suitably revised.

Materials and methods:  Concerns regarding the testing of normality of the distribution of data were clarified.

Results: The inappropriate references included in this paragraph have been removed, and the information has been included in the discussions paragraph.